# Therapeutic correction of ApoER2 splicing in Alzheimer's disease mice using antisense oligonucleotides

Anthony J Hinrich[1], Francine M Jodelka[1], Jennifer L Chang[1], Daniella Brutman[2], Angela M Bruno[3], Clark A Briggs[3], Bryan D James[4], Grace E Stutzmann[3], David A Bennett[4], Steven A Miller[5], Frank Rigo[6], Robert A Marr[3] & Michelle L Hastings[1,*]

## Abstract

Apolipoprotein E receptor 2 (*ApoER2*) is an apolipoprotein E receptor involved in long-term potentiation, learning, and memory. Given its role in cognition and its association with the Alzheimer's disease (AD) risk gene, apoE, *ApoER2* has been proposed to be involved in AD, though a role for the receptor in the disease is not clear. *ApoER2* signaling requires amino acids encoded by alternatively spliced exon 19. Here, we report that the balance of *ApoER2* exon 19 splicing is deregulated in postmortem brain tissue from AD patients and in a transgenic mouse model of AD. To test the role of deregulated *ApoER2* splicing in AD, we designed an antisense oligonucleotide (ASO) that increases exon 19 splicing. Treatment of AD mice with a single dose of ASO corrected ApoER2 splicing for up to 6 months and improved synaptic function and learning and memory. These results reveal an association between *ApoER2* isoform expression and AD, and provide preclinical evidence for the utility of ASOs as a therapeutic approach to mitigate Alzheimer's disease symptoms by improving *ApoER2* exon 19 splicing.

**Keywords** Alzheimer's disease; Antisense oligonucleotides; ApoER2; Splicing; Therapeutics
**Subject Categories** Neuroscience; Pharmacology & Drug Discovery

See also: **CR Wasser & J Herz** (April 2016)

## Introduction

Alzheimer's disease (AD) is a neurodegenerative disorder characterized histologically by amyloid plaques and neurofibrillary tangles and behaviorally by progressive cognitive impairment and learning and memory deficits. Apolipoprotein E (APOE) genotype is the most prevalent known genetic risk factor for late-onset AD (Rebeck *et al*, 1993; Strittmatter & Roses, 1996). Although the mechanism by which ApoE modifies AD risk is not entirely clear, there are apoE isoform-specific effects on synaptic plasticity, cell signaling, lipid transport, and neuroinflammation as well as on amyloid-β (Aβ) peptide aggregation and clearance in the brain (Kanekiyo *et al*, 2014; Kim *et al*, 2014). ApoE interacts with several receptors that are members of the low-density lipoprotein receptor (LDLR) family. These receptors appear to be involved in most aspects of normal apoE function and allele-specific function in AD (Holtzman *et al*, 2012). For this reason, apoE receptors may be promising therapeutic targets for mitigating the effects of apoE and Aβ toxicity in AD.

One member of the LDLR family, apolipoprotein E receptor 2 (*ApoER2* or *LRP8*), elicits cellular effects through its interactions with ApoE as well as Reelin and selenoprotein P (Li *et al*, 2003; He *et al*, 2007; Burk & Hill, 2009; Rogers *et al*, 2011; Holtzman *et al*, 2012). In the brain, ApoER2 has been shown to mediate a signaling cascade that affects long-term potentiation (LTP) and ultimately learning and memory (Herz & Chen, 2006; Wasser *et al*, 2014; Telese *et al*, 2015). Following the binding of Reelin, ApoER2 clusters and induces tyrosine phosphorylation of Dab1, which activates the src family of kinases. ApoER2 isoforms that have the exon 19-encoded domain can subsequently interact with PSD-95 (Trommsdorff *et al*, 1998; Beffert *et al*, 2005; Hoe *et al*, 2006), a postsynaptic density protein important for synapse formation and function (Prange *et al*, 2004). These interactions lead to phosphorylation and activation of NMDA receptors, which increases long-term

1 Department of Cell Biology and Anatomy, Chicago Medical School, Rosalind Franklin University of Medicine and Science, North Chicago, IL, USA
2 Department of Biology, Lake Forest College, Lake Forest, IL, USA
3 Department of Neuroscience, Chicago Medical School, Rosalind Franklin University of Medicine and Science, North Chicago, IL, USA
4 Rush Alzheimer's Disease Center, Rush University Medical Center, Chicago, IL, USA
5 Department of Psychology, College of Health Professions, Rosalind Franklin University of Medicine and Science, North Chicago, IL, USA
6 Ionis Pharmaceuticals, Carlsbad, CA, USA
 *Corresponding author. Tel: +1 847 578 8517; E-mail: michelle.hastings@rosalindfranklin.edu

 

potentiation (LTP). This activity has been shown to counteract the NMDA receptor-dependent synaptic suppression that is induced by amyloid-β (Aβ) peptide (Kamenetz *et al*, 2003; Snyder *et al*, 2005; Durakoglugil *et al*, 2009). In this way, ApoER2 signaling may counteract the toxic effects of amyloid-β peptide in Alzheimer's disease, though a direct link between ApoER2 function and AD has not been definitively shown.

*ApoER2* signaling requires 59 amino acids that are encoded by the alternatively spliced penultimate exon of the gene, documented as exon 19 in mice (Beffert *et al*, 2005) and exon 18 in humans (Clatworthy *et al*, 1999). For simplicity, we will refer henceforth to the alternative exon as exon 19 in both humans and mice. The ApoER2 protein domain encoded by this alternative exon mediates the interaction of ApoER2 with PSD-95 to activate NMDA receptors and also with c-Jun amino-terminal kinase (JNK)-interacting proteins (JIP), which have been implicated in neuronal survival through interactions with JNK (Gotthardt *et al*, 2000; Stockinger *et al*, 2000; Beffert *et al*, 2005, 2006; Hoe *et al*, 2006). ApoER2 protein isoforms that lack exon 19 are associated with defects in long-term memory storage and spatial learning, perhaps through a dominant negative effect on the exon 19-included active isoform (Beffert *et al*, 2005, 2006; Wasser *et al*, 2014). The antagonizing activities of the two alternatively spliced forms of ApoER2 suggest a regulatory role for splicing in signaling (Fig 1A). Given the role of the active form of ApoER2 in memory and learning, alterations in the alternative splicing of this transcript could disrupt fine-tuning at the postsynaptic density and lead to cognitive deficits similar to those seen in AD.

Here, we report a significant decrease in the abundance of *ApoER2* mRNA isoforms that include exon 19, and encode the active form of ApoER2, in autopsy brain tissue samples from AD individuals compared to mild-cognitively (MCI) and non-cognitively impaired (NCI) samples. Mice modeled to have AD exhibit a similar reduction in exon 19 inclusion compared to non-AD animals. We identified SRSF1 as a regulator of exon 19 splicing and show that blocking putative binding sites of this protein with an antisense oligonucleotide (ASO) results in an increase in *ApoER2* mRNA that includes exon 19. We demonstrate that this ASO increases exon 19 inclusion in a transgenic mouse model for AD and also improves synaptic function and behavior deficits in these mice that are associated with neurodegeneration and impaired cognitive function in learning and memory. ASOs are emerging as a promising therapeutic platform for neurodegenerative diseases in humans, and our results suggest that improving *ApoER2* exon 19 splicing using ASOs offers a novel treatment approach to protect cognitive function in AD.

# Results

## A decrease in *ApoER2* exon 19 inclusion is associated with Alzheimer's disease

To test whether changes in *ApoER2* exon 19 splicing are associated with cognitive impairment, samples of the middle temporal region of brains from participants of age 70 years and older in the Religious Orders Study who died with no cognitive impairment (NCI), mild cognitive impairment (MCI), or AD were

obtained from the Rush Alzheimer's Disease Center (Table 1). RNA was isolated from the samples and semiquantitative radiolabeled RT–PCR was performed to analyze exon 19 inclusion in *ApoER2* mRNA. There were significant differences between the groups based on an adjusted ANOVA ($F(2,82) = 6.7$, $P = 0.002$, $\omega^2 = 0.118$) with a significant decrease in exon 19 inclusion in AD samples compared to NCI and MCI samples, which were similar to one another (Fig 1B and C).

Because the NCI and MCI groups did not differ in exon 19 inclusion, we next examined the odds of AD in logistic regression models, adjusting for sex, level of education, and age at death. We found that exon 19 inclusion was associated with a lower odds of AD diagnosis (odds ratio [O.R.] = 0.94, 95% confidence interval [C.I.] = 0.90, 0.98) as indicated by the lower level of inclusion in AD compared to NCI and MCI samples. We next examined the relation of exon 19 splicing to level of global cognition and function in five domains of cognition with linear regression models adjusted for age at death, sex, and education. We found that inclusion of exon 19 was positively correlated with global cognition, and all five domains of cognition: working memory, semantic memory, episodic memory, perceptual memory, and visuospatial ability (Table 2). These results indicate that exon 19 inclusion is an indicator of these cognitive abilities and that greater exon 19 inclusion correlates with higher cognitive abilities. Overall, these results suggest that a decrease in *ApoER2* exon 19 inclusion is strongly associated with AD and/or is an important component of the molecular pathway deregulated in the disease.

## A decrease in *ApoER2* exon 19 inclusion is associated with AD in mice

To further test the relationship between AD and *ApoER2* alternative splicing, we analyzed *ApoER2* RNA from the hippocampus of young (1–2 months) and aged (3–7 months old) mice that harbor the human APP695 cDNA transgene with the Swedish (K670N, M671L) and Indiana (V717F) mutations (TGCRND8; Chishti *et al*, 2001). These mice have cerebral β-amyloid (Aβ) deposition at 3 months of age and have neurite degeneration and significant cognitive impairment by 6 months of age (Chishti *et al*, 2001). RT–PCR analysis of *ApoER2* splicing revealed that inclusion of exon 19 is lower in the TgCRND8 AD mice compared to non-transgenic controls at 5–7 months of age (Fig 1D and E). Interestingly, inclusion of exon 19 was higher in younger mice compared to older mice for both AD and non-transgenic mice. These results reveal an age-related decline in exon 19 inclusion, which is more dramatic in AD mice relative to non-transgenic controls.

## Antisense oligonucleotides that target intronic splicing silencers increase human *ApoER2* exon 19 inclusion

If inclusion of ApoER2 exon 19 decreases in AD, then improving the splicing may be beneficial to disease pathology (Fig 2A). In order to increase exon 19 inclusion, we used antisense oligonucleotides (ASOs) targeted to the intron upstream and downstream of exon 19 with the intent of blocking the activity of putative intronic splicing silencers that are frequently located within the introns flanking alternatively spliced exons (Hua *et al*, 2008; Huelga *et al*, 2012).

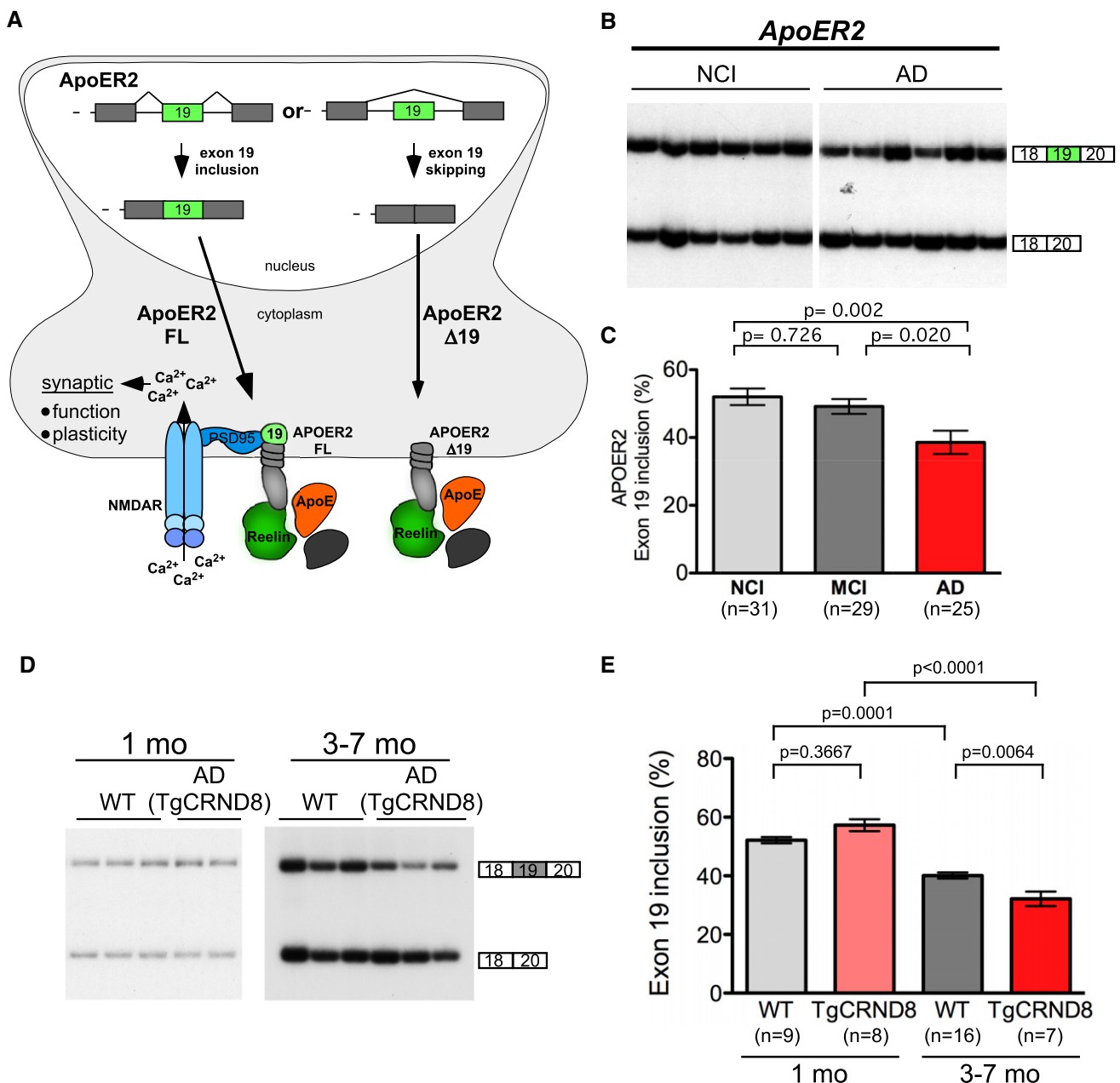

**Figure 1.  *ApoER2* exon 19 inclusion is lower in human and mouse AD brains compared to non-AD samples.**

A   Schematic of ApoER2 expression and signaling via ligand binding (e.g., Reelin, ApoE). ApoER2 alternative splicing of exon 19 produces two ApoER2 protein isoforms with distinct functions. While both isoforms of ApoER2 (+ex19 and Δex19) can bind ligand, a protein domain of ApoER2 encoded by exon 19 interacts with PSD95 to mediate association of ApoER2 with the NMDAR complex following Reelin binding. This signaling leads to calcium influx through NMDARs and the induction of long-term potentiation (LTP).

B   RT–PCR analysis of RNA from mid-temporal autopsy brain sections obtained from NCI, MCI, or AD subjects. Representative gel of results is shown.

C   Quantitation of RT–PCR products of all samples analyzed from postmortem brain tissues (mean ± s.e.m.). An omnibus significant effect was obtained using the one-way analysis of variance (one-way ANOVA) with Tukey posttest to determine the *P*-value. The number of samples analyzed (*n*) is indicated.

D   RT–PCR analysis of RNA from the hippocampus of 1- to 2- and 3- to 7-month-old non-transgenic (WT) or TgCRND8 (AD) mice.

E   Quantification of the relative ratio of exon 19 inclusion for TgCRND8 (AD) and WT mice (mean ± s.e.m.; one-way ANOVA with Tukey posttest). The number of samples analyzed (*n*) is indicated.

Source data are available online for this figure.

The ASOs were 18-mers with 2′-*O*-methoxyethyl (2′MOE) nucleotides. Each ASO was designed to be offset by 5 nts from the previous ASO beginning 90 nts upstream of the 3′ss and continuing to 12 nts upstream of the 3′ss and another set that begins at the 5′ss and continues to 88 nts downstream of the splice site (Fig 2B). ASOs were individually transfected into HeLa cells, and RNA was isolated

**Table 1. Clinical, demographic, and neuropathological characteristics of participant samples.**

| | Clinical diagnosis | | | | Group comparison | Pairwise comparison* |
|---|---|---|---|---|---|---|
| | **NCI (*n* = 31)** | **MCI (*n* = 29)** | **AD (*n* = 25)** | **Total (*n* = 84)** | | |
| Age at death (years) | | | | | | |
| Mean ± SD (Range) | 83.7 ± 6.0 (71–96) | 88.01 ± 6.3 (75–100) | 88.63 ± 6.3 (76–101) | 86.63 ± 6.5 (71–101) | *P* = 0.007[b] | NCI < (MCI, AD) |
| Number (%) female | 63 | 79 | 76 | 71 | *P* = 0.257[a] | |
| Education (years) | | | | | | |
| Mean (±SD) (Range) | 18.4 ± 3.1 (12–25) | 19.2 ± 2.8 (14–25) | 19.3 ± 2.9 (14–26) | 18.9 ± 2.9 (12–26) | *P* = 0.723[b] | |
| PMI (h) | | | | | | |
| Mean ± SD (Range) | 5:18 ± 0.2 (1:00–27:00) | 6:57 ± 0.2 (3:00–17:00) | 7:29 ± 0.3 (1:30–23:20) | 6:38 ± 0.2 | *P* = 0.781[b] | |
| ApoE ε4 allele, Number (%) | 4 (13.3%) | 8 (27.6%) | 9 (36%) | 21 (25%) | *P* = 0.13[a] | |
| Braak score | | | | | | |
| 0 | 0 | 0 | 0 | 0 | | |
| I/II | 13 | 7 | 2 | 22 | *P* < 0.0001[b] | AD < (NCI, MCI) |
| III/IV | 16 | 18 | 12 | 46 | | |
| V/VI | 2 | 4 | 11 | 17 | | |
| NIA-Reagan diagnosis (likelihood of AD) | | | | | | |
| No AD | 0 | 0 | 0 | 0 | | |
| Low | 18 | 10 | 3 | 31 | *P* = 0.0002[b] | AD < NCI |
| Intermediate | 13 | 16 | 15 | 44 | | |
| High | 0 | 3 | 7 | 10 | | |
| MMSE | | | | | | |
| Mean ± SD | 27.7 ± 2.7 | 27.1 ± 2.2 | 19.8 ± 4.8 | 25.1 ± 4.8 | *P* < 0.0001[b] | AD < (NCI, MCI) |

[a]Chi-square test.
[b]Kruskal–Wallis test.
*Dunn's multiple comparison.
NCI, no cognitive impairment; MCI, mild cognitive impairment; AD, Alzheimer's disease; PMI, postmortem interval; MMSE, Mini-Mental State Exam.

**Table 2. Summary of linear regression models examining the relationship of exon 19 inclusion to different measures of cognitive function.**

| Variable | B | SE | *P* |
|---|---|---|---|
| Global cognition | 0.019 | 0.005 | 0.0002 |
| Working memory | 0.012 | 0.005 | 0.0196 |
| Semantic memory | 0.019 | 0.005 | 0.0001 |
| Episodic memory | 0.023 | 0.008 | 0.0033 |
| Perceptual memory | 0.017 | 0.006 | 0.0098 |
| Visuospatial ability | 0.018 | 0.006 | 0.0056 |

Analyses were adjusted for the effects of age, sex, and education.

and analyzed by radioactive semiquantitative RT–PCR. We identified eight contiguous ASOs, ASOs 15–22, that improved exon 19 splicing up to sixfold (Fig 2B). These ASOs define a 53 nucleotide intronic splicing silencer (ISS) located 6 to 59 nucleotides downstream of the 5'ss of exon 19 that presumably block negatively

acting trans-acting protein factors from binding and inhibiting splicing (Fig 2C).

We next analyzed the 53 nucleotide ISS for conserved binding sites for SR proteins using ESEfinder (Cartegni *et al*, 2003; Smith *et al*, 2006). We identified two sequences conserved between humans and mouse that match the consensus motifs for SRSF1. One or both of the motifs are predicted to be blocked by base pairing with the ASOs 15–22, which enhance exon 19 splicing (Fig 2C). These results suggest that ISS sequences downstream of exon 19 limit splicing of the exon and can be blocked to improve splicing of the exon.

To test whether SRSF1 regulates exon 19 splicing, we analyzed *ApoER2* RNA from Hela cells depleted of SRSF1 following treatment with siRNA. Knockdown of *SRSF1* caused a significant increase in exon 19 inclusion (Fig 2D and E). Similarly, knockdown of SRSF1 in a mouse primary cell line derived from adult kidney also resulted in a significant increase in exon 19 inclusion (Fig 2F and G). These results implicate SRSF1 as a negative regulator of *ApoER2* exon 19 splicing and suggest that ASOs that block binding of this protein can increase splicing of the exon.

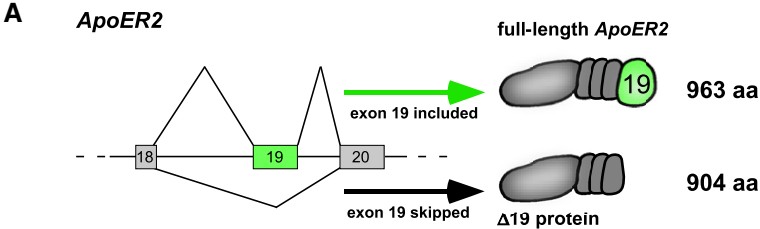

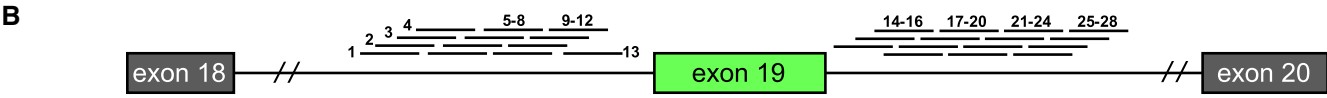

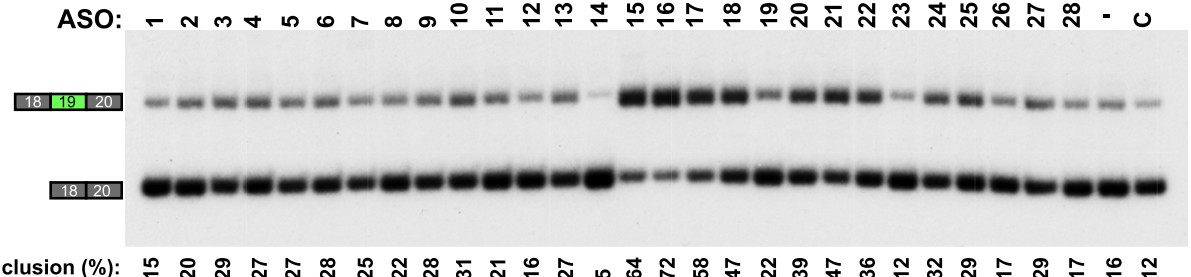

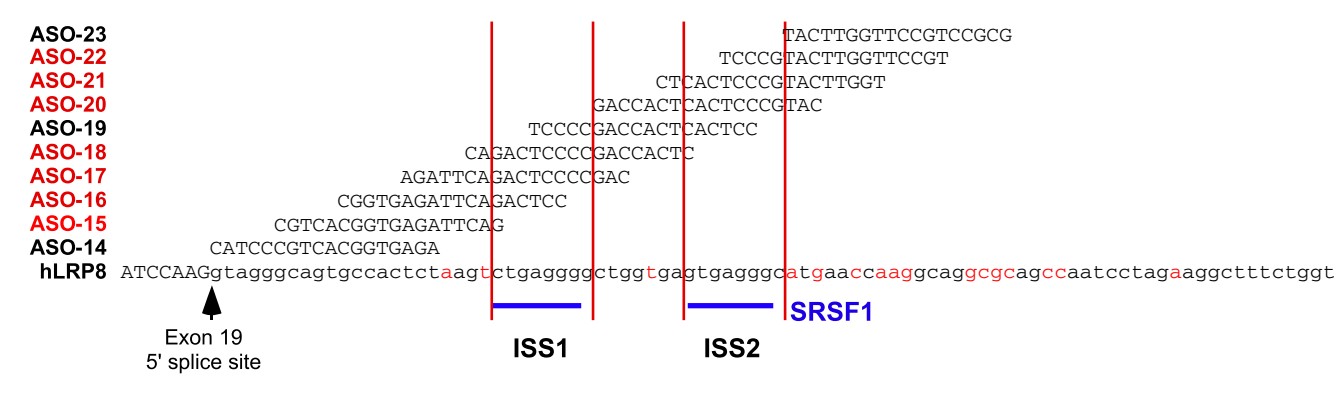

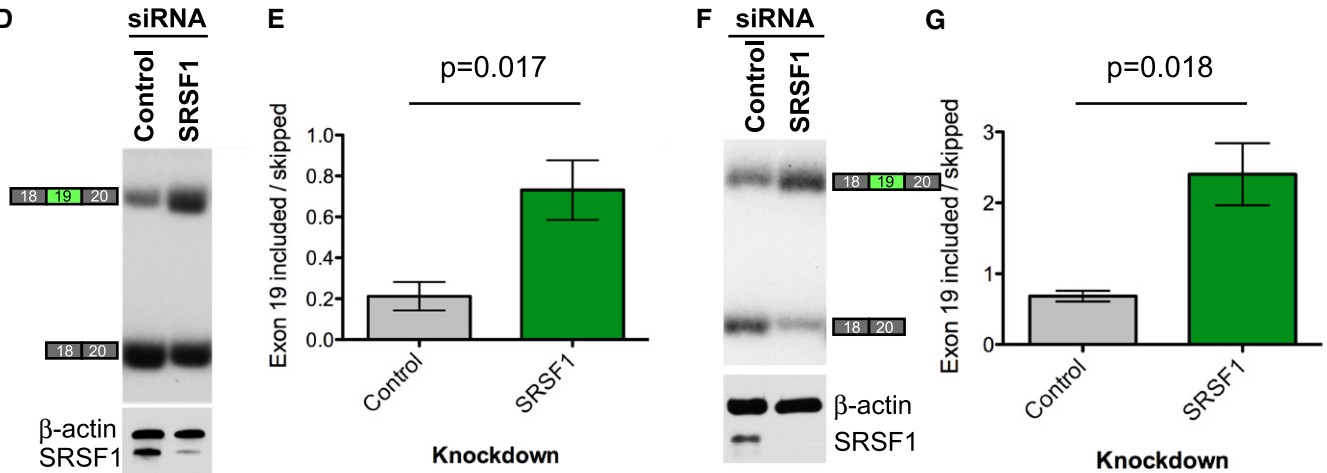

**Figure 2.**

◀

**Figure 2.   Antisense oligonucleotides improve human *ApoER2* exon 19 inclusion.**

A    Gene structure of *ApoER2* exons 18-20 and the resulting RNA and protein products. Boxes are exons and lines represent introns. Diagonal lines represent splicing.
B    Top, diagram of ASOs tested below, mapped to their position of complementarity on *ApoER2*. Bottom, RT–PCR of RNA isolated from HeLa cells transfected with the indicated ASO at a final concentration of 80 nM. RNA spliced forms are labeled. Quantification of the percentage inclusion is shown below the image and calculated as: Inclusion (%) = [exon 19 included/(exon 19 included + exon 19 skipped)] × 100.
C    Sequence and *ApoER2* target region of most active ASOs from (B). Exonic and intronic sequences are in capital and lower case letters, respectively. Conserved nucleotides between human and mouse *ApoER2* are shown in black, and non-conserved nucleotides are in red. Predicted, conserved binding sites for splicing factors are labeled as intronic splicing silencers (ISS1 and ISS2).
D    Top, RT–PCR analysis of *ApoER2* exon 19 splicing in HeLa cells depleted of SRSF1 using an SRSF1-specific siRNA. Control is a non-specific siRNA. Bottom, immunoblot analysis of SRSF1 from HeLa lysates depleted of SRSF1. β-actin was included as a control.
E    Quantification of exon 19 splicing in (D). Error bars show s.e.m. *P*-value was determined using Student's two-tailed *t*-test.
F    Top, RT–PCR analysis of exon 19 splicing in a primary mouse kidney cell line depleted of SRSF1 by RNAi. Control is a non-specific siRNA. Bottom, immunoblot analysis of SRSF1 from HeLa lysates depleted of SRSF1. β-actin was included as a control.
G    Quantification of exon 19 splicing in (E). Error bars show s.e.m. *P*-value was calculated using Student's two-tailed *t*-test.

Source data are available online for this figure.

## Antisense oligonucleotides increase murine *ApoER2* exon 19 inclusion *in vitro* and in transgenic AD mice

In preparation for testing whether ASOs can improve exon 19 inclusion *in vivo*, we first screened for ASOs that increase inclusion of mouse *ApoER2* exon 19 *in vitro*. We designed and tested ASOs that basepair specifically to the intronic region upstream and downstream of exon 19 (Fig 3A). Individual ASOs were transfected into a mouse primary cell line derived from adult kidney and assessed exon 19 splicing by RT–PCR. In contrast to the human ASOs, which only increased exon 19 inclusion when targeted to the intron downstream of exon 19, murine ASOs that bound upstream and downstream of exon 19 improved inclusion (Fig 3A). The intronic splicing silencer region in intron 19 downstream of exon 19 (Fig 2C) is also present in mouse as evidenced by the increase in inclusion observed with ASOs targeted to the region downstream of position +6 relative to the upstream exon, though a more extended intronic region appears to encompass sequences that act as splicing silencers (ASOs 15–28; Fig 3A). The concentration of one of the most active ASOs, ASO-15, that gave half of the maximal effective activity (EC50) was 5 nM (Fig 3B).

To test the efficacy of the ASOs *in vivo*, we administered ASO 14, 15, 16, 21, and 25 to adult, wild-type (WT), non-transgenic mice by intracerebroventricular injection and collected the hippocampus 21 days later (Fig 3C). RT–PCR analysis of ApoER2 exon 19 splicing of RNA collected from the tissue revealed robust stimulation of exon 19 inclusion with a 65% increase in the percent of *ApoER2* mRNA transcripts that include the exon (Fig 3D and E). The ASOs had a similar effect on exon 19 inclusion *in vivo*. ASO-21 was selected for further study because it resulted in the highest mean improvement in inclusion (Fig 3E).

## A single dose of ASO improves exon 19 inclusion for up to 6 months

We next tested whether ASO-21 could increase exon 19 inclusion and ApoER2 expression in a mouse model of human Alzheimer's disease. We used male and female mice of the TgCRND8 line (Janus *et al*, 2000; Chishti *et al*, 2001). These transgenic mice have a rapidly developing amyloid pathology with all mice displaying amyloid plaques in the cortex and hippocampus by 3 months of age that are accompanied by impaired performance on learning

and memory tasks and an increase in locomotor activity (Janus *et al*, 2000; Chishti *et al*, 2001; Walker *et al*, 2011). Because of the rapid disease onset and not knowing *a priori* how early, pre- or post-symptomatically, mice would benefit from an increase in full-length ApoER2, we treated TgCRND8 (AD) mice and non-transgenic (WT) littermates as early as possible, 1 or 2 days after birth by ICV injection either with a non-specific control ASO (ASO-C), which has no cellular target, or with ASO-21 specific for *ApoER2*. ASOs were tolerated well, and no adverse effects on the mice were observed. The weights of the mice treated with ASOs were not significantly different than those that were not treated with ASO (Fig EV1A), and there was no evidence of microgliosis or astrocytosis in the mice treated with ASO-21 (Fig EV1B and C) (Meng *et al*, 2014). ASO treatment also did not affect the general activity level as indicated by the distance they traveled in an open-field area (Fig EV1D). We analyzed exon 19 inclusion in mRNA from the hippocampus of the mice at postnatal day 8 (P8) and at 4 and 6 months of age by RT–PCR analysis. There was a significant increase in exon 19 inclusion as early as 1 week after injection in mice treated with ASO-21 compared to those treated with a control ASO (Fig EV2A). This increase in exon 19 inclusion was maintained at 4 months of age (Fig 4A and B) and up to 6 months of age (Fig EV2B). A corresponding increase in ApoER2 protein isoforms that include the domain encoded by exon 19 was detected by immunoblot analysis using an antibody specific to the exon 19 sequence (Fig 4C and D). The effect of the ASOs on exon 19 inclusion was also observed in the cortex of mice (Fig EV2C). These results demonstrate the long-lasting effect of the ASOs and validate our approach for increasing exon 19 inclusion in the AD mouse model after onset of pathology.

## ApoER2 splice-modulating ASO-21 improves hippocampal basal synaptic transmission without affecting synaptic plasticity or amyloid-β peptide abundance

ApoER2 can modulate synaptic function in an exon 19-dependent manner (Weeber *et al*, 2002; Beffert *et al*, 2005). To evaluate whether the increase in ApoER2 exon 19 inclusion that is induced by ASO-21 treatment could affect synaptic transmission in AD mice, we used field potential electrophysiological recordings to measure synaptic strength and plasticity in the hippocampal Schaffer collateral—CA1 pyramidal neuron pathway of 20-week-old TgCRND8 AD and WT mice. Basal synaptic strength was depressed by about 60%

**Figure 3.  Antisense oligonucleotides improve mouse *ApoER2* exon 19 inclusion.**

A   Top, ASOs mapped onto the mouse *ApoER2* exon 19 gene region. Bottom, RT–PCR of RNA isolated from mouse cells transfected with different ASOs. Quantification of percent exon 19 inclusion is shown below the gel image.

B   ASO improves exon 19 inclusion in a dose-responsive manner. RT–PCR analysis of RNA from mouse cells transfected with increasing doses of ASO-15. The percent inclusion of exon 19 is shown below the gel image. The half-maximal effective concentration (EC$_{50}$) was calculated using GraphPad Prism version 6.0 (GraphPad Software, San Diego, CA) after fitting the data using nonlinear regression with normalized response and variable slope.

C   Sequence and *ApoER2* target region of most active mouse ASOs from (A). Exonic and intronic sequences are in capital and lower case letters, respectively.

D   RT–PCR analysis of RNA isolated from the hippocampus of mice treated as adults by ICV injection of indicated ASOs. C refers to a non-specific control ASO.

E   Quantitation of ApoER2 exon 19 inclusion from (D) (mean ± s.e.m., one-way ANOVA with Bonferroni multiple comparison test).

Source data are available online for this figure.

in ASO-C-treated AD mice, as indicated by the input–output curves (Fig 5A). In AD mice treated with ASO-21, the magnitude of this synaptic transmission deficit was reduced to a level that was not significantly different than WT (Fig 5A). Paired-pulse facilitation (PPF), a measure of short-term presynaptic plasticity, was not different in ASO-C-treated AD compared to WT mice, and ASO-21 treatment did not affect PPF in the AD mice (Fig 5B). Long-term potentiation (LTP) also was not different in ASO-C-treated AD compared to WT mice, and ASO-21 treatment did not affect LTP in

the AD mice (Fig 5C and D). These results indicate a deficit in synaptic strength in AD mice that is ameliorated with ASO-21 treatment.

One characteristic of AD in humans and mouse models of the disease is the presence of an elevated level of amyloid-β (Aβ), which can cause synaptic dysfunction (Shankar & Walsh, 2009). Reelin signaling, potentially through ApoER2, has been shown to counteract this Aβ toxicity (Kamenetz et al, 2003; Snyder et al, 2005; Durako-glugil et al, 2009). To determine whether an increase in ApoER2

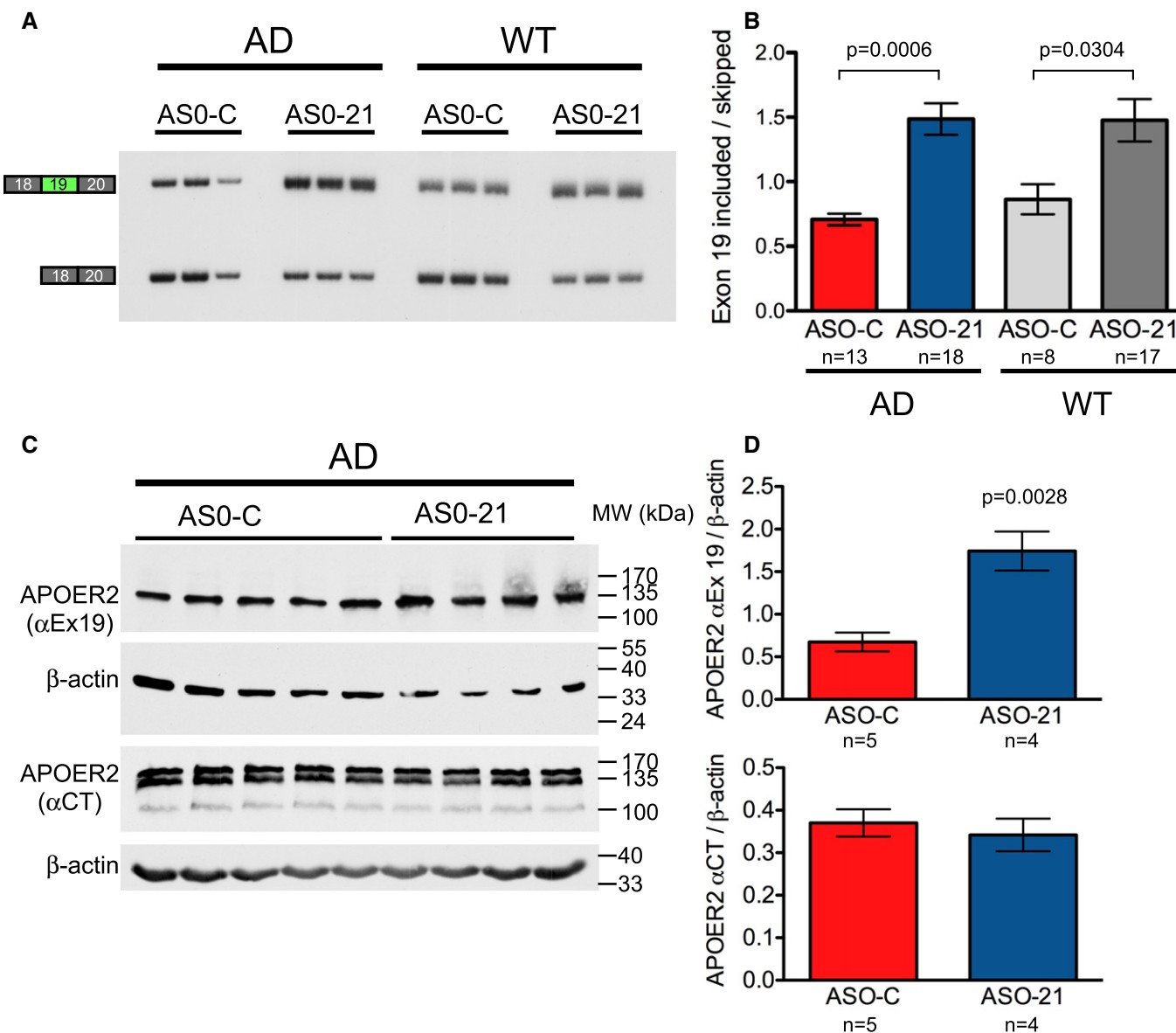

**Figure 4.    ASO-21 improves ApoER2 exon 19 inclusion *in vivo* 4 months post-treatment.**

A    RT–PCR analysis of *ApoER2* exon 19 splicing in RNA isolated from hippocampus of 4-month-old AD and WT mice treated with ASO-21 or ASO-C at P1-2.
B    Quantitation of exon 19 splicing (mean ± s.e.m.; Kruskal–Wallis test with Dunn's multiple comparison test). The number of samples analyzed (*n*) is indicated.
C    Immunoblot for exon 19-containing ApoER2 (top) and total ApoER2 (bottom). β-actin was used as a loading control.
D    Quantification of ApoER2 protein isoforms shown in (C) (mean ± s.e.m., Student's *t*-test). The number of samples analyzed (*n*) is indicated.

Source data are available online for this figure.

isoforms containing the exon 19 region recovers synaptic function via a mechanism that involves Aβ abundance, we quantitated Aβ levels in AD mice treated with ASO-C or ASO-21. We found no significant difference in Aβ levels between the two treatments (Fig EV3).

### ApoER2 splice-modulating ASO-21 improves learning and memory in a mouse model of AD

We next tested whether increasing the splicing of exon 19 could rescue the cognitive phenotype of a transgenic animal model of AD. AD mice have a significant decline in performance in learning and memory tasks by 11 weeks of age (Janus *et al*, 2000; Chishti *et al*, 2001). The Morris water maze (MWM) task is commonly used for the assessment of hippocampal circuitry, as performance in the task has been linked to long-term potentiation and NMDA

receptor function (Morris *et al*, 1986). ApoER2 mice lacking exon 19 (ApoER2Δex19) mice also exhibit a deficit in spatial learning in the MWM task (Beffert *et al*, 2005). Thus, we used a similar MWM task to test whether ASO-21 could improve spatial reference learning and memory in AD mice (Morris, 1984; Janus, 2004; Vorhees & Williams, 2006). AD and WT littermates were treated with ASO-21 or ASO-C at PND1-2 by ICV injection and then tested at 10–12 weeks of age in the MWM task. The mice were tested over a 4-day period for their ability to locate a hidden platform in a pool of water and escape from the water (Fig 6A). Total distance travelled to reach the hidden platform was assessed on each day (Fig 6B). Male AD mice had a significantly greater deficit in the MWM task compared to female mice (Fig EV4), and thus, data were analyzed in a gender-specific manner. There was not a significant difference in the swim speed between any of the groups of mice (Fig EV5A). During the training phase, the swim distance to

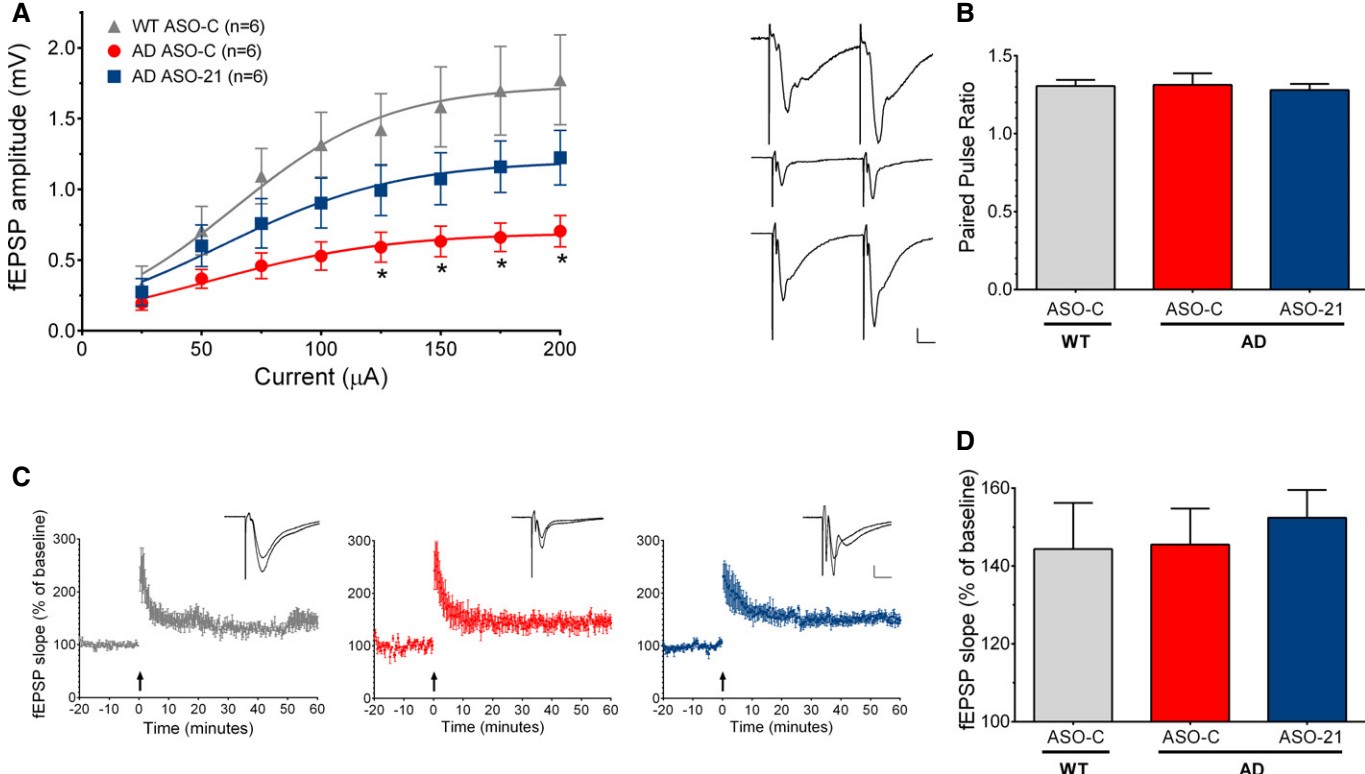

**Figure 5. ASO-21 reduces deficits in basal synaptic transmission strength in the hippocampus of AD mice.**

A   Input–output curves show hippocampal CA1 field excitatory postsynaptic potential (fEPSP) response amplitudes as a function of stimulus intensity delivered to Schaffer collateral afferent fibers. Two-way repeated-measures ANOVA revealed a significant effect of treatment ($F_{(2,119)}$ = 31.62, $P$ < 0.0001) with the Bonferroni multiple comparison test showing significant differences between ASO-C-treated AD and age-matched ASO-C-treated WT (*$P$ < 0.05). Responses from ASO-21-treated AD mice were not significantly different from ASO-C-treated WT or AD mice.

B   Representative paired-pulse facilitation (PPF) responses from ASO-C-treated WT (top), ASO-C-treated AD (middle), and ASO-21-treated AD (bottom) mice are shown to the left. Scale bars = 0.2 mV by 10 ms apply to all traces. Bar graph (right) summarizes the paired-pulse ratios. PPF was not significantly different in AD compared to WT samples, nor was it affected by ASO-21 treatment.

C   LTP was unaffected after ASO-21 treatment and similar in AD compared to WT mice. To combine data from multiple slices, individually measured fEPSP rise slopes were normalized to the average baseline value taken 0–10 min before tetanic stimulation (arrows). The normalized values are plotted as a function of time for ASO-C-treated WT (gray, left), ASO-C-treated AD (red, middle), and ASO-21-treated AD mice (blue, right). Representative traces taken before and 50-60 min after tetanic stimulation are shown in the insets (scale bars = 0.2 mV by 10 ms for all traces).

D   Graph summarizes LTP results as the ratio of mean fEPSP rise slope measured 50–60 min after tetanic stimulation divided by that taken 0–10 min before tetanic stimulation. There was no significant difference among groups. Data information: All data are shown as mean ± s.e.m. and are from the same hippocampal brain slices ($n$ = 6 slices from 4, 3, and 4 WT/ASO-C, AD/ASO-C, and AD/ASO-21 animals per group, respectively).

find the hidden platform progressively decreased in all groups of mice, demonstrating a learning effect (Fig 6B). Both male and female AD mice treated with ASO-C had significantly higher average distances to the platform compared to WT ASO-C-treated mice on training days two-four as expected (Fig 6B and C) (Janus *et al*, 2000; Chishti *et al*, 2001). The male AD mice treated with ASO-21 showed significant improvement in their ability to find the hidden platform compared to male AD mice treated with ASO-C on days two and three of training and the AD mice treated with ASO-21 were not significantly different from the WT mice on day 3 of training. These results demonstrate an improvement in learning related to ASO-21 treatment (Fig 6B and C). ASO-21 did not significantly improve the performance of female AD mice in the MWM task compared to the ASO-C-treated AD mice on any of the days of training, though on day three of training there was a significant difference between the female WT and AD mice treated with ASO-21. For both males and females, there was no significant difference between WT animals treated with ASO-C and those treated with ASO-21 in distance to the hidden platform. Together, these results show that ASO-21 can improve performance of AD mice in the spatial learning tasks associated with the MWM.

Analysis of performance in the probe trials with the platform removed, administered 24 hrs after the final training session, revealed no significant difference in the percent time in the platform quadrant among the groups (Fig EV5B). This lack of an effect has been observed previously in MWM tasks with the TgCRND8 mice and may be due to the fact that all groups of mice have effectively learned the location of the platform by the end of the training and the extinction of the memory does not occur within the 24 hrs between the last trial and the probe test (Janus *et al*, 2000). Indeed, an assessment of the average distance to find the hidden platform on the first trial of each day shows no significant difference in performance between the AD mice treated with ASO-C and ASO-21 by the third and fourth day of training (Fig EV5C).

We examined the correlation between ApoER2 exon 19 inclusion and performance in the MWM in the all groups (AD, non-AD, ASO-C-, and ASO-21-treated) (Fig 6D) or in the transgenic AD animals that were treated with ASO-C or ASO-21 separately (Fig 6E), and found that there was a significant correlation between the percent of ApoER2 mRNA with inclusion of exon 19 and the distance traveled to reach the hidden platform on day 3 of the MWM (Fig 6D and E). Together, these results suggest that an increasing amount of ApoER2 exon 19 inclusion inversely correlates with better performance in learning and memory tasks and demonstrate that the ASO-21-mediated increase in exon 19 inclusion improved spatial memory learning in transgenic AD mice.

## Discussion

Here, we show that *ApoER2* splicing is deregulated in Alzheimer's disease in humans and correction of this defect with antisense oligonucleotides is therapeutic in mice. ApoER2 functions in signaling pathways that are critical for brain development and neuronal maintenance, which, when disrupted, lead to learning and memory deficits similar to those seen in Alzheimer's disease (Beffert *et al*, 2005, 2006; Wasser *et al*, 2014). *ApoER2* expression is regulated, in part, by alternative splicing of exon 19, which codes for a protein domain that is essential for signaling. Here, we demonstrate that the relative abundance of the *ApoER2* isoform that includes exon 19 is lower in the brain of persons with AD compared to persons without cognitive impairment, suggesting that a decrease in active ApoER2 isoforms may be a hallmark of AD. We find a similar decrease in exon 19 inclusion in the brains of mice modeled to have AD. We identified a splicing-related protein that inhibits exon 19 splicing and designed ASOs to block the binding site for this protein. We found that early, pre-symptomatic treatment of AD-like mice with an ASO that increases ApoER2 exon 19 inclusion has no adverse effects on the animals and results in an improved performance at spatial learning tasks compared to mice treated with a control ASO. In all, we have identified a molecular defect associated with AD and developed a treatment that corrects this defect and improves learning in mice modeled to have AD. Our findings are important for understanding and treating AD as well as other neuropsychiatric and neurodegenerative disorders that are associated with reduced signaling through the ApoER2 pathway such as schizophrenia, autism, Down's syndrome, and bipolar disorder (Folsom & Fatemi, 2013).

Alternative splicing of ApoER2 has been previously investigated in the brains of AD individuals, and no differences in exon 19 splicing were observed between non-cognitively impaired and AD brains (Clatworthy *et al*, 1999). While this study was important in establishing the expression repertoire of the gene in human neuronal tissue, the data was assessed in a qualitative rather than quantitative manner and the sample size was small ($n = 3$), and lacked sufficient power to identify a significant effect.

The cause of the decrease in ApoER2 exon 19 inclusion in AD brains is not clear. There are two single nucleotide polymorphisms (SNPs) in the exons flanking exon 19 (rs3737983 (2622T>C) and rs5174 (R952Q)), though neither an association of these SNPs with AD nor an effect on splicing has been clearly demonstrated. Aside from genomic sequence variations, alterations in the expression of proteins involved in alternative splicing may occur in AD and elicit changes in ApoER2 exon 19 inclusion. Indeed, widespread changes

---

**Figure 6.  Antisense oligonucleotides improve learning and memory in a mouse model of AD.**

A   Representative water maze pathway traces of the last four trials of day 3. Trial 1 is shown in red, trial 2 in green, trial 3 in blue, and trial 4 in black.

B   Morris water maze (MWM) analysis of acquisition performance of male (left) and female (right) mice represented as the mean distance (cm) that mice traveled to find the submerged platform plotted per day (mean ± s.e.m.) Symbols represent statistically significant difference within the same genotype (*AD ASO-C vs. AD ASO-21 and #WT ASO-C vs. WT ASO-21) or between groups with same treatment (ᐃAD ASO-C vs. WT ASO-C and ^AD ASO-21 vs. WT ASO-21) on the corresponding day (two-way repeated-measures ANOVA with Tukey multiple comparison test; **$P < 0.01$, ^$P < 0.05$, ᐃᐃᐃᐃ$P < 0.0001$). See source data for statistical results.

C   Morris water maze analysis integrated distance (Zhang *et al*, 2014) (area under the curve, AUC) for male (left) and female (right) WT and AD mice treated with control ASO (ASO-C) or ASO-21 (mean ± s.e.m.; one-way ANOVA, Tukey posttest). The number of samples analyzed ($n$) is indicated.

D, E   Correlation analysis of performance on day 3 of the MWM with exon 19 skipping among all groups of mice (D) or between AD mice treated with ASO-C or ASO-21 (E). Male and female mice are indicated. Pearson correlation coefficient ($r$), covariance ($\sigma$), and two-tailed *P*-value are shown.

Source data are available online for this figure.

**Figure 6.**

in alternative splicing have been reported in AD (Tollervey *et al*, 2011; Twine *et al*, 2011; Bai *et al*, 2013). Regulation of exon 19 alternative splicing is likely a complex process that can be controlled by any number of factors. Our results do suggest that disruption of an SRSF1 binding site in the downstream intron may be the mechanism by which ASO-21 activates exon 19 splicing, indicating that this splicing factor has a role in splicing of the exon. We have no evidence that SRSF1 expression is augmented in AD, and thus cannot conclude that SRSF1 is involved in the deregulated exon 19 splicing that we observe in AD.

The sexual dimorphism in the animals treated with ASO-21 suggests that the treatment is more beneficial to males than females and/or that females may be less severely affected at the age we are testing. Indeed, we observed that female TgCRND8 mice performed better in the MWM task compared to male mice (Fig EV4), suggesting that the females may be refractory to ASO-21 treatment in part because they are not as severely affected as the male mice at the time tested. Gender-related differences in the severity and onset of AD symptoms and pathology have been observed in humans as well as in animal models of the disease (Dubal *et al*, 2012; Mielke *et al*, 2014). In animal studies, numerous factors may play a role in sexual dimorphism, including the impact of the estrous cycle in female mice as well as handling and other variables that may have gender-specific effects.

The effect of the decrease in ApoER2 exon 19 inclusion in AD and the consequent improvement in learning when inclusion is increased in AD mice is likely mediated by ApoER2 ligands such as Reelin and apoE. We hypothesize that ApoER2 deregulation creates an imbalance in the signaling pathway, which may be exacerbated by additional changes in the abundance or form of its ligands. For example, a decrease in Reelin has been shown to be associated with amyloid accumulation, and Reelin overexpression in mice has been shown to improve cognition in mouse models of AD (Chin *et al*, 2007; Kocherhans *et al*, 2010; Herring *et al*, 2012; Cuchillo-Ibanez *et al*, 2013; Krstic *et al*, 2013; Pujadas *et al*, 2014). A recent report has also shown that when Reelin is knocked out in mice, they become sensitive to amyloid-induced synaptic suppression and have learning and memory deficits (Lane-Donovan *et al*, 2015). Models propose that these effects are mediated by a direct interaction between Aβ peptide and Reelin, which limits Reelin signaling through its receptors (Pujadas *et al*, 2014). Increasing the amount of active ApoER2 may ameliorate the effects of a change in ligand availability. Indeed, we found no evidence that ASO-21 treatment altered Aβ abundance in AD mice, suggesting that the improvement in synaptic function associated with the increase in exon 19 splicing is not mediated by a direct reduction in Aβ, per se, but rather may result from increased competitiveness of the exon 19-containing ApoER2 isoform for Reelin and consequently the improved likelihood of forming a functional ApoER2/Reelin signaling complex. Regardless of the exact mechanism, our study demonstrates a link between ApoER2 and AD and validates the receptor as a target for AD therapeutics.

In field potential recordings, basal synaptic strength was reduced in the CA3-CA1 Schaffer collateral pathway in 20-week-old ASO-C-treated TgCRND8 AD mice and this deficit was ameliorated by ASO-21 treatment. The deficit in basal synaptic strength in TgCRND8 mice is consistent with prior electrophysiological studies in TgCRND8 mice (Jolas *et al*, 2002; Arrieta-Cruz *et al*, 2010; Wang *et al*, 2011; Kimura *et al*, 2012) and may be consistent with a recent study demonstrating an overall reduction of hippocampal synaptic spine volume, synaptic markers, and glutamate receptor subunit content in TgCRND8 mice (Sclip *et al*, 2014). We did not observe synaptic plasticity deficits in PPF or LTP in these ASO-C-treated AD mice, indicating that the mechanisms supporting short-term and long-term plasticity are functional at pre- and postsynaptic sites. While some studies have demonstrated LTP deficits in 6- to 12-month-old TgCRND8 mice (Wang *et al*, 2011; Kimura *et al*, 2012), another study found increased LTP in 20-week-old TgCRND8 mice (Jolas *et al*, 2002). Similarly, in Tg2576 mice, another model of β-amyloid deposition, there is evidence for depression in hippocampal basal synaptic transmission while effects on LTP are more controversial (Chapman *et al*, 1999; Fitzjohn *et al*, 2001; Jacobsen *et al*, 2006; Waring *et al*, 2012). Interestingly, ASO-21 treatment was able to ameliorate the depression of basal synaptic strength in TgCRND8 mice without impacting PPF or LTP. Whether this reflects a reduction of synaptic content in TgCRND8 mice, and a restoration with ASO-21 treatment, remains to be determined.

New therapeutic approaches and targets are needed in order to develop an effective treatment for AD. Only a handful of drugs have been approved by the Food and Drug Administration (FDA) for the treatment of the disease, and unfortunately, the effectiveness of these drugs has been marginal (Schneider *et al*, 2014). ASOs are a promising alternative therapeutic platform given their specificity and other favorable pharmacokinetic properties. The ASO that we describe in the present work, ASO-21, could be considered an enhancer of NMDA receptor activity, given the role of ApoER2 on signaling through the receptor via ligands such as ApoE and Reelin. NMDA receptors have been the targets of many drug discovery programs for AD, with both agonists and antagonists of activity pursued. One drug that has been approved for the treatment of AD, memantine, is an NMDA receptor antagonist, though it has had variable effects on memory and cognition in humans and does not slow the progression of the disease (Schneider *et al*, 2014). Enhancers of NMDA receptors, such as glycine, and D-amino acids, such as D-serine, have also been considered for the treatment of AD and have been shown to improve cognitive and behavioral symptoms in other neuropsychiatric disorders such as schizophrenia and post-traumatic stress disorder (Parsons & Ressler, 2013; Schneider *et al*, 2014). Overall, evidence suggests that therapeutic interventions that modulate NMDA neurotransmission may be an effective way to treat behavioral and psychological symptoms of Alzheimer's disease if a potent enough effector can be developed (Huang *et al*, 2012).

Antisense oligonucleotides are emerging as a promising and powerful therapeutic platform. ASOs have many features of an ideal drug, including high specificity and stability, low toxicity, and favorable pharmacokinetics *in vivo* (Havens *et al*, 2013; Lentz *et al*, 2013; Rigo *et al*, 2014). ASO-based therapies are being used in the clinic for the treatment of a number of diseases and are in clinical trials for several others including an ASO for the treatment of the pediatric neurodegenerative disease, spinal muscular atrophy (Kole *et al*, 2012). ASOs may be particularly powerful as drugs in the central nervous system given their long half-life, with effects lasting up to 6 months in the brain following a single injection (Hua *et al*, 2011; Havens *et al*, 2013; Lagier-Tourenne *et al*, 2013;

Arechavala-Gomeza et al, 2014; Meng et al, 2014; Rigo et al, 2014). ASOs are also beginning to be explored as a potential therapeutic platform for AD (DeVos & Miller, 2013). Our results suggest that ASOs may offer a promising new drug platform for developing new treatment strategies in AD.

# Materials and Methods

### Source of clinical data and postmortem biospecimens

Data and biospecimens came from participants in the Rush Alzheimer's Disease Center's Religious Orders Study. Briefly, this cohort study funded by the National Institute on Aging began enrollment in 1994 and has a rolling admission with more than 1,200 participants at the time of these analyses. Participants without dementia at baseline agree to annual detailed clinical evaluation and brain donation at death. Details of the study have been reported previously (Bennett et al, 2012). All participants signed an informed consent, an Anatomic Gift Act, and agreed to place their data in a Repository for data and biospecimen sharing. Both the parent study and the Repository were approved by the Institutional Review Board of the Rush University Medical Center. The clinical evaluation included 21, a battery of cognitive performance tests used to inform on clinical diagnoses of dementia and its causes such as Alzheimer's disease, and mild cognitive impairment (MCI) as previously described (Bennett et al, 2002, 2006). A subset of battery of 19 tests was used to assess global cognition, and five cognitive domains including episodic memory, working memory, semantic memory, perceptual speed, and visuospatial ability (Wilson et al, 2002). After death, a summary diagnosis of AD, MCI, and no cognitive impairment (NCI) was made by a neurologist following review of all clinical data without access to neuropathologic data. Each brain underwent a uniform structured neuropathologic assessment that the density of neuritic plaques according to CERAD, Braak stages (neurofibrillary pathology in the neocortex), and NIA-Reagan Institute neuropathologic diagnosis of the likelihood of AD, as previously described (Bennett et al, 2005). Postmortem diagnoses were made without access to clinical data by a neuropathologist. Data and biospecimens used from this cohort included 31 NCI, 29 MCI, and 25 AD cases. They were examined for ApoER2 exon 19 inclusion. Analyses using clinical and postmortem data were performed using SAS 9.3.

### Antisense oligonucleotides

All ASOs were uniformly modified with 2′-O-(2-methoxy) ethyl sugars (2′MOE), phosphorothioate backbone, and 5′-methyl cytosine as described (Baker et al, 1997). ASOs were dissolved in 0.9% saline. ASO sequences are provided in Appendix Table S1. A BLAST search for ASO-21 target sequences revealed no other perfect sequence matches within the mouse genome. Genomic targets with one mismatch to ASO-21 were analyzed for alterations in splicing pattern (when target is within an intron) or protein expression (when target is near the promoter) by RT–PCR or immunoblot, respectively (Fig EV6). A BLAST search with ASO-C revealed no perfect matches to genomic sequences.

### Mice

The TgCRND8 mice harboring the human APP695 cDNA transgene with the Swedish (K670M, N671L) and Indiana (V717F) mutations were used in this study (Janus et al, 2000; Chishti et al, 2001). Age-matched non-transgenic (Non-Tg/WT) mice, based on the same background strain, were used as controls. Both male and female mice of approximately equal numbers were used. Tg and nTg (wild-type) CRND8 breeder mice were obtained from Dr. David Westaway (University of Toronto), and the mice were maintained on a hybrid genetic background (C3H/C57BL/6) (Chishti et al, 2001). The TgCRND8 mice exhibit progressive plaque pathology beginning at 3 months (Chishti et al, 2001). Animals were used and cared for in accordance with NIH guidelines and protocols reviewed and approved by the Rosalind Franklin University's Institutional Animal Care and Use Committee. TgCRND8 mice were genotyped using ear tissue and Red Extract-N-Amp (Sigma, St. Louis, MO). PCR was performed with the primers APP_F and APP_R (see Appendix Table S1 for sequences).

### Cell culture and transfection

HeLa cells were grown in HyClone DMEM/High Glucose media supplemented with 10% FBS. Mouse primary kidney cell line, 68J or 208EE (Lentz et al, 2013), was established from an adult C57BL/6 mouse kidney. For RNAi experiments, HeLa cells were transfected with siRNA (50 nM final concentrations) using Invitrogen Lipofectamine 2000 Reagent (Life Technologies, Carlsbad, CA) according to the manufacturer protocol and as described in previous studies (Wee et al, 2014). Scrambled AllStar siRNA (Qiagen) was used as a control. Cells were grown for 48 h post-transfection, at which point cells were transfected with an additional 50 nM siRNA. Cells were then split 1:2, 24 h after the second RNAi treatment, and total RNA and protein were collected 24 h later. siRNA sequences are provided in Appendix Table S1. For ASO experiment, ASOs (80 nM final concentration) were transfected into human HeLa or mouse kidney-derived 208EE cells. RNA was collected 48 h post-transfection.

### RNA isolation and analysis

RNA was isolated from tissue and cells in culture using TRIzol reagent (Life Technologies, Carlsbad, CA) according to the manufacturer's protocol. For human tissue, RNA was isolated and treated with 4 μg of DNase-I (RNase-free) (Life Technologies) followed by reverse transcription with Superscript III Reverse Transcriptase (Life Technologies) to produce cDNA. For murine tissue, RNA was reverse-transcribed using GoScript reverse transcription system (Promega, Madison, WI). Radiolabeled PCR was carried out as previously described (Lentz et al, 2013), using primers specific for human ApoER2/LRP8 exons 17 and 19: hsLRP8_ex17_For:; hsLRP8_ex19_Rev: or for mouse: musLRP8_ex18_For:; musLR-P8_ex20_Rev (Appendix Table S1). Real-time PCR was performed on an Applied Biosystems (ABI) ViiA™ 7 Real-Time PCR System (through the Molecular Quantification Laboratory at RFUMS) and Taqman assay kits for Aif1, GFAP, and β-actin (Mm00479862_g1, Mm01253033_m1, and 4352933E, respectively, Life Technologies).

Results were analyzed with the $\Delta\Delta C_T$ method (Livak & Schmittgen, 2001).

## Intracerebroventricular injections

For adult mouse studies, transgenic and non-transgenic mice were treated with a single 500 μg dose of either ASO-C or ApoER2 splice-modulating ASOs via ICV injection under isoflurane anesthesia. Each mouse was individually placed in an anesthesia induction chamber until reflex inhibition was verified with a foot or tail pinch. The cranium of the mouse was then stabilized in a stereotaxic apparatus with 18-degree ear bars and tooth bar, such that the bregma and lambda sutures aligned in the same *z* plane. Anesthesia was maintained during surgery with 2% isoflurane delivered by nose cone. The fur on the scalp was clipped and the skin sterilized with 70% ethanol. An incision approximately 1 cm long was made in the scalp along the saggital suture to expose bregma. A hole was drilled in the cranium at a position 0.2 mm posterior and 1.0 mm right of bregma, and a syringe needle was lowered to a depth of 3.0 mm. A total volume of 4.5 μl was injected at a rate of 0.4 μl/min. Tissues were collected 4 weeks post-treatment. Brain tissue isolation was performed on ice-cold PBS. The hippocampus was dissected and either snap-frozen in liquid nitrogen or placed in 1 ml TRIzol prior to RNA extraction. RNA was extracted from tissues via TRIzol reagent (Life Technologies), and radioactive RT–PCR was performed using primers LRP8_ex18F and LRP8_ex20R. Products were separated on a 6% non-denaturing polyacrylamide gel and quantified using a Typhoon 9400 phosphorimager (GE Healthcare).

For neonatal ICV injection, pups (P1-P2) were treated with a single 15 μg dose of either ASO-C or ASO-21 via ICV injection according to a published procedure (Hua & Krainer, 2012). In brief, ASOs were diluted in sterile 0.9% saline, with 0.01% Fast Green FCF and 2.5 μl was injected into the left ventricle using a 33-gauge needle affixed to a glass Hamilton syringe approximately 2.5 mm anterior to the lambda suture and 1 mm lateral to the saggital suture to a depth of 2 mm.

## Protein extraction and immunoblot analysis

Brains were dissected and immediately frozen in liquid nitrogen. Proteins were obtained from frozen tissue by homogenizing tissue in a modified RIPA buffer (Jodelka *et al*, 2010) using a power homogenizer. Cells were lysed with Laemmli buffer. Proteins were boiled in sample buffer and separated on 8% SDS–PAGE and then transferred to PVDF membranes (Millipore). Membranes were blocked in 5% nonfat dry milk in 1× TBST and incubated with ApoER2 C-terminal antibody ab108208 (1:2,000) (Abcam), ApoER2 exon 19-specific antibody (1:1,000) (a gift from Joachim Herz) (Beffert *et al*, 2005), GFAP antibody (1:5,000) (ab7260, Abcam), SRSF1-specific antibody (1:100)(a gift from Adrian Krainer), MetAP2-specific antibody (1:200) (D3I1H, Cell Signalling), or β-actin-specific antibody (1:2,000) (Sigma-Aldrich) followed by incubation with anti-rabbit or anti-mouse HRP-conjugated antibody (1:5,000) (Thermo). Bands were visualized using ECL reagents (Millipore) and by exposing on autoradiography film. Blots were quantified using Image J software v1.48s (NIH).

## Hippocampal slice preparation

Transverse hippocampal slices were prepared from 20-week-old male and female mice in accordance with protocols approved by the Institutional Animal Care and Use Committee at Rosalind Franklin University. Mice were anesthetized with halothane and decapitated. The brains were extracted rapidly, and 400-μm-thick transverse hippocampal slices were cut with a vibrating microtome (Campden Instruments) in ice-cold sucrose cutting solution gassed with 95% $O_2$/5% $CO_2$ and containing (in mM) 200 sucrose, 1.5 KCl, 1.0 $KH_2PO_4$, 0.5 $CaCl_2$, 4.0 $MgCl_2$, 25 $NaHCO_3$, 10 sodium ascorbate, and 20 dextrose. Slices were then immediately transferred to artificial cerebral spinal fluid (aCSF) gassed with 95% $O_2$/5% $CO_2$ and containing (in mM) 130 NaCl, 2.5 KCl, 1.25 $KH_2PO_4$, 2.0 $CaCl_2$, 1.2 $MgSO_4$, 25 $NaHCO_3$, and 10 dextrose. Slices were maintained at 32°C for at least 1 h before use.

## Extracellular field potential recordings

Slices were transferred to a submerged-slice recording chamber (BT-1-13B, Cell MicroControls, Norfolk VA), continuously perfused with oxygenated aCSF (1.5 ml/min) at room temperature (23°C), and visualized using an upright microscope (Olympus BX51WI) and transillumination. All recordings were conducted in hippocampus CA1. Schaffer collaterals were stimulated using a bipolar stainless steel electrode (FHC inc., Bowdoin ME) and 100-μs constant-current pulses delivered from a digitally triggered stimulus isolation unit (A.M.P.I. ISO-Flex, Jerusalem, Israel). Field excitatory postsynaptic potentials (fEPSPs) were recorded in stratum radiatum using a blunt-tip glass micropipette filled with aCSF (1–5 MΩ). Signals were captured at 10 kHz with an Axoclamp 2B amplifier, Digidata 1440A digitizer, and pClamp 10.2 software (Molecular Devices, Sunnyvale CA).

Stimuli were delivered to the Schaffer collaterals at 0.05 Hz. Input–output curves were generated by varying stimulus intensity from 25 to 200 μA in increments of 25 μA. Following that, the stimulus intensity was adjusted for each slice to elicit fEPSP of 40% maximal amplitude for the synaptic plasticity protocols. Paired-pulse facilitation (PPF) was measured using a 50-ms interstimulus interval. Three response pairs acquired at 20-s intervals were averaged, and PPF was quantified as the ratio of the second fEPSP amplitude to the first. For long-term potentiation (LTP), after establishing a 20-min baseline at 0.05 Hz, LTP was induced at the same stimulus intensity by high-frequency tetanic stimulation (100 Hz for 1 s delivered twice with a 10-s interval) and quantified by comparing the average fEPSP rise slope measured 50–60 min post-tetanization to that measured 0–10 min pre-tetanization.

## Aβ enzyme-linked immunosorbent assay (ELISA)

Mouse brains were dissected in 1X phosphate-buffered saline (PBS) divided into halves. Hippocampi and cortex sections were extracted and snap-frozen. Frozen cortex sections were homogenized in RIPA buffer (10% sodium deoxycholate, 10% SDS, 0.5M EDTA, protease inhibitor cocktail in 1X PBS). Samples were then diluted 1:10 in 5M guanidine-HCl and allowed to rotate overnight at room temperature. Samples were diluted 1:10 again in a Tris buffer solution (50 mM

Tris–HCl pH = 8, 0.03 % Tween-10, protease inhibitor cocktail) and centrifuged at 16,000 $g$ for 20 min at 4°C. Supernatants were collected and diluted (1:20) for Aβ analysis. Aβ levels were measured using a Human/Rat β-amyloid (42) ELISA Kit (Wako Chemicals, Richmond VA, USA). The ELISA experiments were performed as per manufacturer's instructions. Protein levels were measured via Bradford Protein Assay Kit (Pierce) and used to normalize Aβ levels measured.

## Behavioral testing

Behavioral studies were performed on TgCRND8 (AD) and non-transgenic (WT) littermates that were not treated, or treated at P1 or P2 by ICV injection with a control ASO or the ApoER2-specific ASO-21. Individuals performing behavioral tests were blinded to the animal treatment/genotype status. Mice from AD and WT groups were littermates (i.e., ASO-C-treated litters contained both WT and AD mice). Treatment of litters was chosen at random, and each group had subjects from multiple litters from different parents. Power analysis using preliminary results from the third day of testing was performed to estimate sample size that would be adequate (80% power), given the estimated effect size, to achieve statistical significance (α = 0.05) compared to controls.

### Open field
Test groups were 68- to 80-day-old mice comprised of WT, ASO-C, WT, ASO-21, AD, ASO-C, and AD, ASO-21. Untreated WT and AD were also tested. Mice were placed individually in an open 54 × 54 cm square white acrylic chamber with 20-cm-high walls and allowed to explore freely for 10 min. This process was performed once daily over three consecutive days. On the third day, path traces of the mouse were recorded in the open-field chamber during the exploration period using VideoMot software and activity was quantitated.

### Morris water maze
Test groups were 77- to 85-day-old mice. Testing was conducted in a circular pool (48 inch/1.2 meter diameter) filled to a depth of 26 inches (66 cm) with 23°C water made opaque with gothic white, non-toxic, liquid tempera paint in a room with prominent extra-maze cues at least 16 inches (41 cm) from the pool edge. Four unique proximal cues were affixed to the 8-cm-high interior pool wall above water level at 0, 90, 180, and 270°. Mice were placed in one of four starting locations at 0, 90, 180, and 270° facing the pool wall and allowed to swim until coming to rest atop a 4-inch (10 cm) square Plexiglass platform submerged under 0.5 cm water, or until a maximum of 60 s. Upon finding the platform, mice were left on the platform for 20 s before reentry at the next start point or removal to the home cage. If mice did not find the platform within 60 s, they were guided to the platform by the experimenter and remained on the platform for 20 s before removal. Trials were performed once at each start point per session. The experimenter was blind to the genotype of the mice. Latency to reach the platform, distance traveled to reach the platform, swim speed, time spent in each of 4 quadrants, and time spent along the walls were obtained using automated video tracking software (HSVimage). Mice were tested for four consecutive days with eight trials per day, four trials in the AM and four trials in the PM with a 3-h period

### The paper explained

#### Problem
Alzheimer's disease (AD) is the most common form of dementia and is characterized by progressive loss of memory and other intellectual and behavioral abilities. There is no effective treatment for AD, despite major drug development efforts. Thus, there is a need for new approaches to detect and prevent the advancement of the disease. Hallmarks of the disease include memory loss and behavioral abnormalities indicative of loss of synaptic function and plasticity. The apoE receptor, ApoER2, has been shown to mediate synaptic signaling and may be an effective intervention point to ameliorate the learning and memory deficits observed in AD. There are two forms of ApoER2 produced by alternative splicing of the pre-mRNA. One isoform is active and the other is inactive in synaptic signaling. Increasing the levels of the active form of ApoER2 may be therapeutic in AD.

#### Results
To determine whether ApoER2 may be a good target for drug development in AD, we first assessed whether the balance of the active and inactive forms of the molecule was disrupted in the brains of humans that had succumbed to the disease. We found a significant reduction in the active form of ApoER2 in the brains of AD patients relative to samples from unaffected individuals. We also observed this imbalance in ApoER2 in a mouse model of AD, indicating that ApoER2 isoform expression is deregulated in the disease and may be a marker of AD. We next developed a therapeutic approach to correct the balance of ApoER2 isoform expression using antisense oligonucleotides (ASOs). ASOs are being used for the treatment of other neurological diseases and thus offer a promising therapeutic approach in AD. We identified ASOs that target ApoER2 alternative splicing and effectively increase production of the active ApoER2 isoform. We found that AD mice treated with a single dose of ASO had a long-lasting improvement in the levels of the active ApoER2 isoform, improved synaptic function and performance on learning and memory tasks. Our results identify a deregulated signaling pathway in AD that, when corrected using a therapeutic dose of ASO, can ameliorate deficits associated with the disease.

#### Impact
These findings suggest that the ApoER2 signaling pathway could serve as a marker for AD as well as a drug target for AD therapy. Antisense oligonucleotides are a promising drug platform that offers an effective and long-lasting approach to reversing the deficits in ApoER2 splicing associated with the disease.

between the AM and PM sessions. Twenty-four hours after the final training day, mice were tested for the time spent in the probe quadrant during a 60-s time period when the platform was removed. One week after the MWM test, all mice were tested in six trials (two blocks of three trials followed with a 2-h rest period in between) for their ability to find a cued platform. Only those that had an average latency to reach the cued platform of < 60 s were used in the final MWM analysis.

## Quantification and statistical analysis

Statistical analyses were performed as described in the text and Figure legends using SPSS v.21, GraphPad Prism 5 or 6, or SAS 9.3. All experiments will be designed to ensure sufficient power (80%) to detect departures from the null hypothesis. Variability analyses

from preliminary experiments were used to guide the selection of sample sizes. All data are shown as mean ± s.e.m. When using one-way analysis of variance (ANOVA) or *t*-tests, data sets were checked for normality with the Shapiro–Wilks normality test and Bartlett's (ANOVA) or Levene's F (*t*-test) test of homogeneity of variance to determine whether parametric tests ($P > 0.05$) or nonparametric tests should be used. Independent samples *t*-tests (parametric) were used to determine significant differences between two groups. One-way analysis of variance (parametric analysis) or Kruskal–Wallis test (nonparametric analysis) was used to compare three or more groups. Correction for multiple comparisons was performed as indicated in the text. Significance for *t*-tests and analyses of variances are reported with a two-tailed test with an α-level of 0.05.

Expanded View for this article is available online.

## Acknowledgements

We thank Jeffrey Huang, Megan Miller, Mallory Havens, Rida Khan, Paige Keasler, and Jennifer Akamine for technical assistance, reagents, and valuable comments on the manuscript and Joachim Herz for the APOER2(αEx19) anti-body. The work was supported by grants from NIH NS069759 to MLH and the Midwest Proteome Center grants NIH S10 OD010662, and NIH grants P30AG10161 and R01AG15819 to DAB.

## Author contributions

FMJ, BDJ, DAB, AMB, and RAM developed, provided samples, produced, and analyzed data from the human sample experiments. MLH and FR designed and analyzed antisense oligonucleotides. FMJ, AJH, and DB performed and analyzed *in vitro* studies. AJH, FMJ, JLC, and RAM designed, performed, and analyzed results from animal studies. CAB and GES developed and performed the electrophysiological experiments and analyzed the data. SAM, DAB, BDJ, and MLH performed statistical analysis. MLH developed the study, conceived the experimental plans, and analyzed the data. MLH, AJH, RAM, and CAB wrote the manuscript. All authors read and edited the manuscript and gave final approval of the manuscript version to be published.

## Conflict of interest

F.R. is a paid employee of Ionis Pharmaceuticals. M.L.H. and F.R. have patents pending with the United States Patent and Trademark Office for the ASOs and the targeting approach.

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
