## [Review Process File · EMBO Molecular Medicine]

Therapeutic Correction of ApoER2 Splicing in Alzheimer's Disease Mice Using Antisense Oligonucleotides

Anthony J. Hinrich, Francine M. Jodelka, Jennifer L. Chang, Daniella Brutman, Angela M. Bruno, Clark A. Briggs, Bryan D. James, Grace E. Stutzmann, David A. Bennett, Steven A. Miller, Frank Rigo, Robert A. Marr, Michelle L. Hastings

Corresponding author: Michelle Hastings, Rosalind Franklin University of Medicine and Science, North Chicago, USA

Review timeline:

Submission date:	10 September 2015
Editorial Decision:	13 November 2015
Revision received:	20 December 2015
Editorial Decision:	18 January 2016
Revision received:	25 January 2016
Accepted:	27 January 2016

Transaction Report:

Editor: Céline Carret

1st Editorial Decision

13 November 2015

Thank you for the submission of your manuscript to EMBO Molecular Medicine and for your patience while the manuscript was under review. We have heard back from two out of the three referees whom we asked to evaluate your manuscript.

While reviewers 1 and 2 delivered their evaluations in a timely manner indeed, we still have not received the other reviewer's input. As the evaluations from the first two reviewers are consistent and a further delay cannot be justified any longer, I have decided to proceed based on these evaluations.

In the next 10 days, should we receive the other review and only if it raises significant caveats, these would need to be taken into consideration. We would not, however, ask you to comply with any further-reaching requests.

You will see that both reviewers are generally supportive of your work and underline its considerable potential interest. However, referee 2 makes good suggestions in how to improve the data even more and we would like to give you the opportunity to perform these experiments and overall address all issues prior to resubmit for further consideration.

Please note that it is EMBO Molecular Medicine policy to allow only a single round of revision and that, as acceptance or rejection of the manuscript will depend on another round of review, your responses should be as complete as possible.

Please read below for important editorial formatting that should be addressed in order to proceed quickly should the article re-evaluation be positive.

I look forward to receiving your revised manuscript.

***** Reviewer's comments *****

Referee #1 (Comments on Novelty/Model System):

Well-done and controlled study on the impact of altering Apoer2 ex19 splicing, a novel target for AD therapy. Study shows potentially high therapeutic impact, using the optimal preclinical mouse model for this study, the mouse.

Referee #1 (Remarks):

Excellent, high impact study. Well done and well controlled. No additional experiments are needed.

A few minor issues to be fixed in the text during revision:

Page 3, second to last line: This [NMDA receptor phosphorylation inducing] activity [of ApoER2] has been shown to counteract the NMDA receptor-dependent synaptic suppression.... The appropriate reference after 'suppression' is missing. That would be 'Durakoglugil et al., PNAS 106, 15938-15943 (2009)'

The authors refer to the alternatively spliced exon in the ApoER2 intracellular domain as 'exon 19', which is correct when referring to the mouse. In humans, it is exon 18 (Clatworthy et al., Neuroscience 90, 903-911 (1999)). Please cite this reference as well and discuss the discrepancy (Clatworthy found no change in splicing in AD brains - why? Insufficient sample size?).

Page 8, line 6 from bottom: Sentence grammatically wrong. Looks like the word 'mice' is duplicated.

Referee #2 (Comments on Novelty/Model System):

I felt that the technical quality of the data, novelty, and medical impact were all high and I do not have any alternative suggestions, other than those noted in my comments to the authors.

I think the choice of model systems was excellent: initial screening of antisense oligos was done with cell lines and the subsequent biological relevance of those was tested with AD mouse model

Referee #2 (Remarks):

Hinrich et al., 'Therapeutic correction of deregulated ApoER2 splicing in Alzheimer's disease'
 In this exciting study, Hinrich and co-authors investigate the mechanism related to deregulated splicing of ApoER2 receptor, which is involved in ApoE signaling, cognitive learning and memory through activation of NMDA receptors that increase long-term potentiation. Using post-mortem autopsy samples, the authors show that an isoform of ApoER2 lacking exon 19 is prevalently expressed in AD patients as opposed to persons with no-cognitive impairment. Through screening of antisense oligonucleotides (ASO) targeting intronic regions surrounding the exon 19, the authors successfully demonstrate that it is possible to improve the inclusion of exon 19 by blocking the binding sites of putative intronic splicing silencers, which they show to binds splicing factor SRSF1.

Based on these preliminary experiments with cultured cells the authors continue with mouse model of AD. They show that intracerebroventricular injections of the most efficient ASO leads to an increased inclusion of exon 19 both in wt and AD mouse. Using the Morris water maze, they show that there was a significant improvement in learning in AD mice injected with the ASO, albeit in a gender-specific manner. Importantly, the effect on exon 19 inclusion is stable at least 6 months, which supports the therapeutic potential of ASOs. In sum, this is a carefully conceived and well-written study showing the potential of ASOs in treating AD and other similar disorders.

However, the authors need to address the following issues in a revised manuscript:

1. Since one of the main characteristics of AD is accumulation of amyloid plaques and neurofibrillary tangles, the authors should indicate whether the ASO treatment had any effect on amyloid plaque formation in the AD mouse model. This data has a significant impact on the therapeutic potential of the ASOs and also understanding the mechanistic basis of improved learning and memory functions, regardless whether the result would be positive (inhibition of plaque formation) or negative (no effect on plaque formation).
2. The study would have benefited from the inclusion of long-term potentiation studies in the mice studied, to determine if these were actually altered in a correlative manner with the inclusion/exclusion of exon 19 in ApoER2.
3. Page 7, paragraph 3.... Figure 2a should be 3a.
4. Page 5: Awkward wording in the second last sentence, last paragraph "more splicing" should rather refer to increased inclusion of this exon.
5. Page 10: last three sentence of first paragraph: I found this part confusing and difficult to follow. In particularly, the final sentence seems to contradict the previous sentences - unless the authors clearly state the gender specific effects, which I think is the root of this confusion.
6. Figure 5d: Would have benefited from the inclusion of WT correlation data too and maybe even more days post training (if feasible).
7. Page 11: "We identified several splicing-related proteins..." only SRSF1 is mentioned in the text.
8. Figure 3b: It seems that EC50 data is derived from a single experiment (error bars missing). Also, it seems that the curve in the figure is more artistic than an actual fit to a dose effect curve. This should be corrected in the revised manuscript.
9. There were no legends for the supplementary figures.
10. The authors should discuss the potential off-target effects of the ASO used in this study.

Point-by-Point Response to Referees

Referee #1 (Remarks):

Excellent, high impact study. Well done and well controlled. No additional experiments are needed.

A few minor issues to be fixed in the text during revision:

Page 3, second to last line: This [NMDA receptor phosphorylation inducing] activity [of ApoER2] has been shown to counteract the NMDA receptor-dependent synaptic suppression.... The appropriate reference after 'suppression' is missing. That would be 'Durakoglugil et al., PNAS 106, 15938-15943 (2009)'

Response:

We agree. This reference has now been added after this sentence.

The authors refer to the alternatively spliced exon in the ApoER2 intracellular domain as 'exon 19', which is correct when referring to the mouse. In humans, it is exon 18 (Clatworthy et al., Neuroscience 90, 903-911 (1999)). Please cite this reference as well and discuss the discrepancy (Clatworthy found no change in splicing in AD brains - why? Insufficient sample size?).

Response:

These are both very good suggestions. We have added a sentence in the introduction to clarify the nomenclature of exon 18/19 in humans and mice. For simplicity and clarity, we prefer to adopt a common naming of the exon between species and thus refer to both human and mouse forms as exon 19 throughout the manuscript, since this is the name by which it is most often referred. Because additional exons have been annotated in the gene, the historical numerical exon names are somewhat ambiguous. Nonetheless, we maintain the original names but also indicate that the alternative exon of interest is the penultimate exon for both mouse and humans for easy identification.

We have now included a paragraph in the discussion to address this previous study. Clatworthy et al did not quantitate their results and indeed, if they had, the sample size (n=3) would be insufficient to establish a significant difference according to our analysis.

Page 8, line 6 from bottom: Sentence grammatically wrong. Looks like the word 'mice' is duplicated.

Response:

We have corrected this typographical error.

Referee #2 (Remarks):

In sum, this is a carefully conceived and well-written study showing the potential of ASOs in treating AD and other similar disorders.

However, the authors need to address the following issues in a revised manuscript:

1. Since one of the main characteristics of AD is accumulation of amyloid plaques and neurofibrillary tangles, the authors should indicate whether the ASO treatment had any effect on amyloid plaque formation in the AD mouse model. This data has a significant impact on the therapeutic potential of the ASOs and also understanding the mechanistic basis of improved learning and memory functions, regardless whether the result would be positive (inhibition of plaque formation) or negative (no effect on plaque formation).

Response:

We agree that it would be appropriate to assess whether correcting ApoER2 splicing and the cognitive deficits in AD mice results in any change in amyloid status, which may contribute to the behavior changes in our ASO-21-treated AD mice. Thus, we have made every effort to address this question within a reasonable time-frame and in a manner that fits with the scope of the current study. To this end, we have now analyzed amyloid- β peptide abundance in AD mice that were treated with ASO-C or ASO-21 and show that there is no change in A β 42 levels at 4 months of age in the mice. This result suggests that the effect of correcting ApoER2 splicing on synaptic function and learning and memory is not mediated by a change in toxic A β . These results are presented as Supplemental Fig S3.

We did not include an analysis of amyloid plaque formation. We did perform these experiments on a small subset of animals at 4 months of age. We saw no difference in plaque abundance. However, we only had two AD mice treated with ASO-21 and three AD mice treated with ASO-C (four sections each) available at the time, too small of a number with which to test the significance of the results. The impetus for expecting that ApoER2 signaling would affect plaques is based, in part, on reports that one of its ligands, Reelin, can affect A β abundance and plaque formation (Kocherhans et al., 2010; Pujadas et al., 2014). This effect has been proposed to be a result of a direct interaction of the ligand with A β peptide rather than an effect on the downstream signaling pathway. Since, ApoER2 signaling is predicted to occur downstream of this Reelin/A β interaction, we predict that the increase in synaptic function observed with ASO-21 treatment is likely due to an increase in ApoER2 signaling but would not necessarily be expected to affect Reelin/ A β interactions or plaque abundance. We agree that it could be helpful to prove this hypothesis. However, we note that these same studies have shown that the effect of Reelin on plaque formation was a late event, with significant plaque reduction occurring only at 12 months of age (Pujadas et al., 2014). Because we see cognitive improvement at 10-12 weeks of age, we predict that plaque abundance may not be a factor in the recovered function. In all, given the extensive time to acquire results, which we expect to be negative, we hope that the reviewer will agree that this experiment is outside the scope of the present work. We have expanded the discussion on this topic as well.

2. The study would have benefited from the inclusion of long-term potentiation studies in the mice studied, to determine if these were actually altered in a correlative manner with the inclusion/exclusion of exon 19 in ApoER2.

Response:

The Referee has a very good point here. We have now included a new Figure (Fig 5), which includes results from an electrophysiology study of brain slices from mice to determine the effect of the ASO-mediated increase in exon 19 inclusion on synaptic function and LTP. Field potential recordings showed that basal synaptic strength was reduced in the CA3-CA1 Schaffer collateral pathway in 4 month-old ASO-C-treated AD mice and this deficit was ameliorated by ASO-21 treatment. We did not observe synaptic plasticity deficits in paired pulse facilitation or LTP in the AD mice, which, as discussed in the revised manuscript, is consistent with previous findings that LTP is not affected until 6-12 months of age in TgCRND8 mice.

3. Page 7, paragraph 3.... Figure 2a should be 3a.

Response:

We have corrected this error.

4. Page 5: Awkward wording in the second last sentence, last paragraph "more splicing" should rather refer to increased inclusion of this exon.

Response:

We have reworded this sentence as suggested.

5. Page 10: last three sentence of first paragraph: I found this part confusing and difficult to follow. In particular, the final sentence seems to contradict the previous sentences - unless the authors clearly state the gender specific effects, which I think is the root of this confusion.

Response:

We have changed the second to last sentence, which had an incorrectly placed "not", and included a referral to the female mice for clarity.

6. Figure 5d: Would have benefited from the inclusion of WT correlation data too and maybe even more days post training (if feasible).

Response:

We agree that this is a good suggestion that will strengthen our conclusions. We have added a graph (new Fig 6D) with the correlation analysis between all the groups of animals tested in the MWM. We agree with the author that this is an

important addition as it supports the premise of the study, that ApoER2 exon 19 inclusion is linked to improved cognitive function. We have also retained the original graph (new Fig 6E) in order to have an isolated comparison of the ASO effect in AD mice.

7. Page 11: "We identified several splicing-related proteins..." only SRSF1 is mentioned in the text.

Response:

We have corrected this sentence to accurately reflect the data presented for SRSF1.

8. Figure 3b: It seems that EC50 data is derived from a single experiment (error bars missing). Also, it seems that the curve in the figure is more artistic than an actual fit to a dose effect curve. This should be corrected in the revised manuscript.

Response:

The half-maximal effective concentration (EC₅₀) was calculated using GraphPad Prism version 6.0 or higher (GraphPad Software, San Diego, CA) after fitting the data using nonlinear regression with normalized response and variable slope. Given the number of concentrations tested, a curve can be fit quite well, as shown. We have included the 95% CI of the data points based on the curve and included additional description in the legend to make the analytical method more clear.

9. There were no legends for the supplementary figures.

Response:

We have corrected this oversight and the Supplementary Figure legends are included in this version.

10. The authors should discuss the potential off-target effects of the ASO used in this study.

Response:

We have addressed potential off-target effects in the Materials and Methods under the Antisense Oligonucleotide section. We have also included a new Supplemental Figure (Fig S6) that lists genomic sites with only one mismatch from ASO-21. Two of these potential target sites fall deep within intronic regions, far from splice sites. This location is unlikely to encompass splicing signals and thus ASO-21 is not predicted to have an impact on splicing at these sites were it to form a stable interaction. Nonetheless, we have assessed the splicing of the exons flanking these potential off-target sites and found no evidence of aberrant splicing or changes in alternative splicing (Fig S6b). One of the potential off-

target binding sites falls upstream of an ORF, which could affect gene expression. For this case, we have performed immunoblot analysis to demonstrate that ASO-21 does not affect expression of this target (Fig S6c).

Thank you for the submission of your revised manuscript to EMBO Molecular Medicine. We have now received the enclosed reports from the referee who was asked to re-assess it. As you will see the reviewer is now globally supportive and I am pleased to inform you that we will be able to accept your manuscript pending final editorial amendments.

Please submit your revised manuscript within two weeks. I look forward to seeing a revised form of your manuscript as soon as possible.

***** Reviewer's comments *****

Referee #2 (Comments on Novelty/Model System):

Well-done and controlled study on the impact of altering Apoer2 ex19 splicing, a novel target for AD therapy. Study shows potentially high therapeutic impact, using the optimal preclinical mouse model for this study, the mouse.

Referee #2 (Remarks):

The authors have done excellent work in revising the manuscript. They have addressed all of my concerns.

Corresponding Author Name: Michelle L. Hastings
 Journal Submitted to: EMBO Molecular Medicine
 Manuscript Number: 05846